# Direct Observation of Evolution from Amorphous Phase to Strain Glass

**DOI:** 10.3390/ma15227900

**Published:** 2022-11-09

**Authors:** Andong Xiao, Zhijian Zhou, Yu Qian, Xu Wang

**Affiliations:** Frontier Institute of Science and Technology and State Key Laboratory for Mechanical Behavior of Materials, Xi’an Jiaotong University, Xi’an 710049, China

**Keywords:** Fe-Pd, amorphous alloy, strain glass, crystallization, ferromagnetic

## Abstract

The amorphous phase and strain glass are both disordered states of solids. The amorphous phase is an atomic packing disordered phase, while strain glass is a glassy state with transformation strain disorder in a crystalline matrix, which both bring extraordinary properties to alloys. Previous studies have mostly focused on the properties and structure of single glass; however, the link between them has seldom been considered. In this work, the specimen of the almost amorphous state was obtained from the heavy-defects-doping Fe_67.8_Pd_32.2_ strain glass ingot by arc melting and 90% cold rolling, which were characterized by amorphous packages in X-ray diffraction and amorphous rings in transmission electron microscope diffraction. The evolution from the amorphous phase (metallic glass) back to strain glass was directly observed by an in situ high-resolution transmission electron microscope, which revealed that strain nanodomains began to form on the amorphous matrix below the crystallization temperature of the amorphous phase. Here, direct observation of the evolution process provides a theoretical basis for achieving precise control of crystallinity to obtain the desired microstructure, while the study of the unusual crystallization process offers a possible way to tailor the mechanical and functional properties through tuning the amorphous and strain glass coexistence. This work presents the specific pathway and realization possibilities for the design of glass composite materials with enhanced properties.

## 1. Introduction

Alloys exhibit different glassy or crystalline states under various conditions. Recently, metallic glass and strain glass, as two special alloy states, have attracted much interest due to their extraordinary properties [1,2]. Owing to the disorder of atomic packing, amorphous alloys exhibit high strength, large elasticity, and excellent corrosion resistance [3,4]. Metallic glass is mainly applied as a high-performing functional structural material that combines multiple properties in advanced manufacturing, aerospace, medical, and other fields [4,5]. Fe-based amorphous materials also have potential applications in permanent-magnet micro-devices and data storage media [6]. However, the drawback of limited plasticity restricts the application of metallic glass [5,7]. The crystallization of metallic glass can solve this problem [5,8] and can become a feasible way to design new functional materials [5,9,10].

Many efforts have been devoted to investigating the crystallization of the amorphous matrix [7,11,12,13,14]. Recent research also indicates that a special-strain short-range ordered but long-range disordered state (strain glass) can exist after the crystallization of the amorphous phase [15,16]. Strain glass usually occurs in shape memory alloys when the martensitic transformations are suppressed by heavy structural defects [17,18]. It has been proven that the strain glass state can be formed by introducing point defects, dislocations, nanoprecipitates, etc., into martensite [17]. Furthermore, the strain glass state can bring the materials novel properties such as the Elinvar effect, high damping, and superelasticity over wide temperature ranges [18,19,20].

It has been reported that the strain glass state can be achieved by crystallization from the amorphous state through annealing treatment [15,21]. Amorphization of numerous alloys by severe plastic deformation on the bulk arc-melted ingots has been investigated, while most deformation limitations are restricted by the plasticity and cracking [20]. In the last decade, related work on the effects of plastic deformation and thermomechanical treatment has been conducted using techniques such as cold rolling [6,22,23,24,25], cold drawing [26,27], high-pressure torsion [16,28], and equal channel angular extrusion [29,30]. They offer the possibility to achieve the transformation between metallic glass and strain glass by cold rolling and annealing. However, the lack of direct observation on the evolution of the two kinds of glass makes the transformation path still unclear, so it is difficult to control the degree of recrystallization and the composite structure by controlling the annealing temperature. Among the various strain glass systems that have been reported, such as TiNi-based strain glass [31], TiNb-based strain glass [32], and FePd strain glass [20], FePd strain glass is a good choice due to its ease of processing and feasibility of becoming amorphous by plastic deformation, while TiNi-based strain glass is too difficult to cold-deform, and TiNb strain glass cannot form an amorphous state through cold rolling. Here, Fe_67.8_Pd_32.2_ alloys, which have been found as strain glass [20], are cold-rolled up to 90% to create enough large amorphous areas. We can achieve our main objectives of directly observing the complete process of evolution from the amorphous phase to the strain glass state by an in situ high-resolution transmission electron microscope (TEM), and establishing the link between the two kinds of glass in the severe cold-rolled FePd. The changes in diffraction pattern and structure at the microscopic level in TEM observations and the changes in viscoelasticity at the macroscopic level by dynamic mechanical analysis together demonstrate this evolutionary process. This investigation introduces new insight into the relationship between the two kinds of glass separately in the noncrystalline matrix and crystalline matrix, which is different from the conventional process from the amorphous phase to crystal after annealing. This work also provides theoretical support and realization possibilities for achieving precise control of the crystallization degree to obtain the desired microstructure of the glass composite to design the desired properties.

## 2. Experimental Procedure

Fe_67.8_Pd_32.2_ alloys were prepared from highly pure metal (>99.95 at.%) by arc-melting (SYJ DHL-500 Argon Arc-melting Furnace) under an argon atmosphere at a charge current of 180 A. The ingots were solution-treated at 1373 K for 24 h in evacuated quartz tubes with Ar atmosphere to eliminate compositional nonuniformities, followed by water quenching. After heat treatment, the specimens underwent cold rolling with thickness reduction *ε*_p_ (*ε*_p_ = (*t*_0_ − *t*)/*t*_0_, where *t*_0_ and *t* are the thicknesses before and after cold rolling, respectively) using a double-high mill (JSTZX ZL 42.5-5-1). In the rolling process, the roll was maintained at a speed of 400 r/min and the roll reduction was 0.1~0.6 mm each time according to the sample thickness. When thickness reduction *ε*_p_ was larger than 5 mm, the reduction was 0.6 mm each time. When *ε*_p_ was between 2 mm and 5 mm, the reduction was 0.3 mm each time. When *ε*_p_ was below 2 mm, the reduction was 0.1 mm each time. The specimens were then cut into suitable shapes for different experiments (3 × 3 × 1 mm^3^ for differential scanning calorimeter (DSC), 3 × 1 × 30 mm^3^ for dynamic mechanical analysis (DMA), and 6 × 8 × 1 mm^3^ for X-ray diffraction (XRD)) using wire electrical discharge machining (EDM). Then, they were annealed at 300 K, 400 K, 500 K, and 630 K for 1 h in evacuated quartz tubes with Ar atmosphere, followed by air cooling. All the specimens were etched with HF:HNO_3_:H_2_O 1:4:5 (in volume) solution to remove oxidized surface layers.

In situ X-ray diffraction (Shimadzu 7000 XRD) measurement was used to identify the possible structural change and the extent of amorphization. The latent heat of crystallization transformation was obtained from heat flow in a differential scanning calorimeter (DSC-Q200 from TA Company, Boston, MA, USA) curve with a heating rate of 20 K/min between 297 K and 730 K. The dynamic mechanical properties were performed by Dynamic Mechanical Analysis (DMA-Q850 from TA Company) using the 17.5 mm three-point bending mode with an amplitude of 15 μm (AC field frequency from 0.2 to 20 Hz). The temperature was chosen from 323 K to 123 K with a cooling rate of 2 K/min. The microstructure was observed by transmission electron microscopy (TEM) using a JEM-2100F microscope with a heating holder (Gatan 652). The specimens for TEM were spark-cut from the bulk samples, mechanically polished to 50 mm, and finally electropolished to perforation using a twin-jet electropolisher with CH_3_OH:HClO_4_ 4:1 (in volume) solution at a charge current of 15 mA and constant voltage of 25 V for 30 s.

## 3. Results and Discussion

### 3.1. Amorphous Characterization of Cold-Rolled Alloys

The X-ray diffraction (XRD) results of the initial specimens with different rolling degrees at room temperature are shown in Figure 1. Before cold rolling, the specimen has a face-centered cubic (FCC) phase, and this can be identified from the (111) and (220) diffraction peaks shown in Figure 1(a1,a2). With the increase in the degree of cold rolling, the intensity of the two peaks becomes weaker, while the width of peaks becomes larger, which indicates that the crystalline size decreases and the crystalline structure is gradually destroyed by severe deformation. Furthermore, when *ε*_p_ reaches 90%, as shown in Figure 1(d2), only a low and broad (111) peak is observed, while the (220) peak almost disappears. The sample can, thus, be called an amorphous package. As a result, the transition from an FCC structure crystalline matrix to an amorphous matrix takes place during the rolling process, as illustrated by the schematic diagram of Figure 1e.

To check if most areas of the 90% rolled FePd alloy transforms to the amorphous phase, the microscopic structure and its diffraction pattern are performed below. The bright-field image shows that there is no distinct grain boundary or grain phase in the rolled specimen (Figure 2a). When concentrating on the middle of the area, the selected-area diffraction pattern (SADP) has stronger diffraction halos and a few diffraction-like spots, which can confirm that the sample is an almost amorphous state. To clarify more the structure information represented by the TEM diffraction pattern, this SADP is compared with standard diffraction halos, originating from the simulated diffraction spots of a standard Fe_3_Pd crystal using the Crystal Maker program in Figure 2b. It is noted that the diffraction rings are around the precise position of the spacing of {222}, {440}, and {444} lattice planes instead of inside. It can be attributed that the crystal structure and lattice parameter change during the rolling process. The angle of the peak in XRD and the distance from the diffraction spot to the central spot in TEM diffraction can both reflect the lattice parameters in accordance with the Bragg formula, where the SADP results can also be correlated with the previous XRD results in Figure 1. The (111) and (200) XRD peaks correspond to diffraction spots and amorphous rings at the positions of {222} and {400} lattice planes, which both represent the same crystal faces, and the difference in their values comes from the difference in the way of calibration. The crystal faces calibrated in XRD are based on the Miller index obtained from the unit cell, while TEM diffraction is calibrated by simulated diffraction spots, which are the indices of the crystal face obtained from the primitive cell. In the nonsimple lattice such as a Fe_3_Pd crystal with a face-centered cubic phase or a face-centered tetragonal phase, it is also possible to draw a crystal face through the face center, in which case the face spacing of a primitive cell is only half of that in the unit cell [33]. Therefore, the indices of the crystal face are twice as large as the Miller index in our specimen. Both the {222} TEM diffraction ring and the broadened (111) XRD peak prove that most areas of the 90% rolled FePd specimen transform to the amorphous state. The diffraction-like spots on the ring and the not-completely disappeared XRD peak indicate that a small part of the crystalline phase remains in the rolled alloy, while the near-disappearance of the (220) XRD peak corresponds to the almost invisible {440} amorphous halos.

To further trace the crystallization process, the DSC results of the 90% rolled specimen with twice heating are presented in a stack fashion with a normalized scale, as shown in Figure 2c. Only the first heating DSC curve displays a heat latent peak upon heating, while there is no change in the second heating process. The phenomenon indicates that the FePd alloy transforms from the amorphous phase to the crystal phase in the first heating. However, it does not show any other change in the second heating after finishing crystallization. The glass transition temperature (T_g_~580 K) can be obtained in the first heating DSC curve. The latent heat of crystallization is less than that of general metallic glass fabricated by rapid solidification because the disorder degree of our specimen is lower than the rapid solidification ones and there are still a few nanocrystalline phases in the rolled amorphous phase. More importantly, the heat flow curve already has an undulation below the recrystallization temperature, which indicates that the sample has absorbed heat and undergone some changes. It is extremely different from the normal amorphous phase [5]. It also suggests that the evolution of recrystallization may start below the usually defined recrystallization temperature T_g_.

Based on the above results, it can be concluded that the 90% cold-rolled FePd alloy almost becomes the amorphous phase and the crystallization temperature of the amorphous phase is around 580 K.

### 3.2. Evolution from Amorphous Phase to Strain Glass State

Dynamic mechanical analysis (DMA) methods have been used to analyze the viscoelasticity of materials [34]. The viscoelasticity of materials is sensitive to changes in structure; therefore, DMA can be used to test for small changes in the structure of materials, and in this paper, the results of the phase change tested using DMA can effectively reveal the changes in structure during crystallization.

Unlike FePd with a strain glass phase transition before cold rolling [18], the crystalline order is destroyed after 90% rolling and the specimen only exhibits a normal modulus rise upon the cooling process, while the modulus softening of STG disappears, as shown in Figure 3a. The rising trend in the cooling modulus of the sample becomes flattened when annealed at 400 K, and a slight modulus softening occurs around 280 K, corresponding to a slight peak in Tan Delta. A slight jittering of the DSC heat flow around 400 K has already occurred in Figure 2c, and the change in the DMA results of the specimen annealed at 400 K in comparison with that of the initial one also indicates that the rolled FePd has started to absorb heat and undergo evolution, upon being heated to this temperature. This modulus softening is more pronounced in the samples tempered at 500 K, corresponding to a more intense jittering of the heat flow on warming to 500 K. The dithering of the modulus and loss may originate from different sub-stable regions with varying crystallinity and lattice constants during the crystallization process when the specimen is heated to these temperatures lower than the recrystallization temperature T_g_ obtained from DSC. The significant change in viscoelasticity indicates that the transition from the FePd amorphous phase to the crystalline phase has started below T_g_. For the rolled specimen annealed at 630 K for 1 h above the recrystallization temperature, it again exhibits a typical strain glass transition, manifested by the frequency dependence of the elastic modulus at the strain-glass transition temperature, as shown in Figure 3d. The frequency dependence obeys a Vogel–Fulcher relation ω = ω_0_exp[−E_a_/k_B_(T_STG_ − T_0_)], where ω is the frequency, ω_0_ is the frequency pre-factor, E_a_ is the activation energy, k_B_ is the Boltzmann constant, and T_0_ is the ideal freezing temperature, ~165 K, as shown in the inset of Figure 3d. Thus, the above evidence reveals that the recrystallization process is the evolution of the material from the amorphous phase to strain glass and this evolution already starts below the recrystallization temperature T_g_. This consequence is also consistent with the previous conjecture from DSC results.

In order to study the isothermal crystalline kinetics, the microscopic TEM micrographs of the rolled samples heated at different temperatures (300 K, 400 K, 500 K, 580 K, and 630 K) are provided in Figure 4a–e. The microstructure of the original rolled specimen shown in Figure 4(a2) reveals no obvious grain boundary or grain phase, which means that the sample is almost amorphous. When the specimen is heated to 400 K and 500 K below the crystallization temperature (T_g_), some obscure grains appear of sizes of about 40~70 nm in Figure 4(c2). They are not considered to be crystalline grains, because the main SADPs are still amorphous rings in Figure 4(b3,c3). However, the trend of crystallization occurring below the crystallization temperature can be verified from the increasingly sharp diffraction spots (pointed by yellow arrows). This is also consistent with the previous DMA results. When the specimen is heated to 580 K under the crystallization temperature, obscure grains disappear, while some lamellar nanocrystals and nanodomains appear instead. In order to obtain a clearer nanoscale microstructure, the specimen is annealed at 630 K for 1 h. The obvious grain boundaries indicated by the white and gray lines are likely to divide the specimen into numerous areas with irregular shapes shown in Figure 4(e2) and the partially enlarged inset. Instead of amorphous rings, diffraction spots appear in the SADP image (Figure 4(e3)). More details of the specimen where the white and grey lines divide the specimen (Figure 4(e2)) annealed at 680 K for 1 h are shown in Figure 5. The microstructure of recrystallized samples can be seen more clearly and analyzed more specifically.

### 3.3. Structure Analysis of Recrystallized Strain Glass State

In the high-resolution image (Figure 5a) of the recrystallized sample after annealing, Fast Fourier transformation (FFT) is performed for three regions with significantly different morphology and contrast by the Digital Micrograph program. The obtained FFT diffraction patterns in Figure 5(a1,a3) indicate that the crystal structures of these two regions are different. There are no diffraction spots in the FFT diffraction patterns in Figure 5(a2), so region a2 still maintains the amorphous state. According to the different diffraction patterns, it can be concluded that the specimen is divided into many different crystalline matrices and small amorphous matrices. The previous diffraction patterns are placed here in Figure 5b to determine crystal types of the recrystallized alloy. However, the diffraction patterns of nanocrystals are difficult to analyze due to the too small a grain size and different orientations. In order to further study the crystal structure after recrystallization, the nearby regions with similar orientation around [110] are selected in Figure 6.

The high-resolution TEM (HRTEM) micrographs of the specimen annealed at 630 K for 1 h are shown with the FFT diffraction patterns of several regions along the [110] zone axis in Figure 6. Compared with the standard [110] diffraction spots, the diffraction spots of the {222} plane group are around the standard positions, but diffraction spots of the {004} plane group vary in a large range in Figure 6b. It has been reported that the martensitic transition path of FePd alloy is from a face-centered cubic (FCC) austenite phase to a face-centered tetragonal (FCT) martensitic phase and, thus, its crystal parameters change from *a* = *b* = *c* = 3.818 Å to *a* = *b* > *c*. That means that it is easy to change the crystal parameter *c*, which might account for the fact that diffraction spots of the {004} plane group have various changes. Figure 6c is a schematic picture of a general Fe_3_Pd crystal’s {110} lattice plane generated by the Crystal Maker program and the spacing between the atomic distances can be calculated in the FCC crystal. Similarly, the lattice parameters can be calculated backward based on the atomic spacing, which can be measured in HRTEM images using the Digital Micrograph program. The crystal lattice in region a1 is the same as the standard data, so that is the FCC phase in Figure 6(a1). The crystal lattice in region a2 is different from the standard one. The parameters *a = b* do not change but *c* varies from 3.8 Å to 4.35 Å. From the data in Figure 6(a3), the crystal parameters all change in region a3. The details of the atomic spacing and lattice parameters are listed in Table 1. 

In consequence, careful analysis of the crystallization process by in situ HRTEM indicates the emergence of a strain glass state at a suitable isothermal temperature. As shown in Figure 6d, on the one hand, the nanocrystal or nanoglass acts as intermediate products at the initial crystallization state [27]. On the other hand, it proves the evolution from the amorphous phase to the strain glass state in the heavy-defects-doping composition. This heavy defects-doping, which induces the strain glass state in FePd, may still exist during the rolling process as a nucleation-like core, making the crystallization process easier. Thus, it starts to occur before the crystallization temperature. This is different from previously reported martensite products or the austenite phase after crystallization in low-defects-doping composition [15,21]. The synergy of the thermodynamic driving force and kinetic slow down determines the product of amorphous crystallization. Annealing causes dislocation dissolution and grain growth, and the amorphous phase with the highest disorder extent can crystallize directly into a relatively long-range ordered nanocrystalline phase (or crystalline phases, if the time is long enough and the temperature is high enough) in the case of annealing temperature above crystallization temperature. At this point, a sufficiently large thermodynamic driving force can prevail over the kinetic slow down due to defects so that the martensitic transformation occurs rather than traps a strain glass state [35]. The reduction in annealing temperature mitigates the effects of plastic deformation and, thus, enables a higher level of defects, which causes the system to remain in a state where the kinetic energy is below the local energy barrier after amorphous crystallization [31]; therefore, the system is converted to a strain glass state. Although it is possible to undergo an amorphous-strain glass-crystalline process by short-time high-temperature annealing and long-time low-temperature annealing can eventually crystallize to the crystalline state according to the temperature–time equivalence principle, low-temperature annealing relaxes this process to make it achievable in material preparation and experimental observation, which is the reason why a series of low-temperature annealing is chosen in this paper.

## 4. Conclusions

An unusual crystallization from the amorphous state to the strain glass is investigated by the direct evidence of in situ TEM images and dynamic mechanical analysis in a severely 90% cold-rolled Fe_67.8_Pd_32.2_ alloy. During the in situ annealing process of the cold-rolled amorphous phase, clear diffraction spots, grain boundaries, and nanocrystals (representing crystalline state) and the frequency dependence of the elastic modulus (representing strain glass state) appear above T_g_. This is similar to the recrystallization process of general amorphous alloy. However, unusually, below the crystallization temperature T_g_, there also appears the softening of the modulus with the gradual sharpening of diffraction spots and the appearance of nano-sized obscure grains. These facts reveal that the evolutionary crystallization trend starts below T_g_. The unusual crystallization originates from the fact that the heavy defects-doping in strain glass that remains after rolling reduces the energy required for nucleation in the crystallization process and facilitates the crystallization. In detail, these defects act as the nuclei and grow to form nanocrystals gradually. This work provides a coexistence evolutionary process that is different from the conventional amorphous phase recrystallization, and gives a new route for designing novel glass composite materials with great mechanical and functional properties through tuning the amorphous and strain glass coexistence. 

## Figures and Tables

**Figure 1 materials-15-07900-f001:**
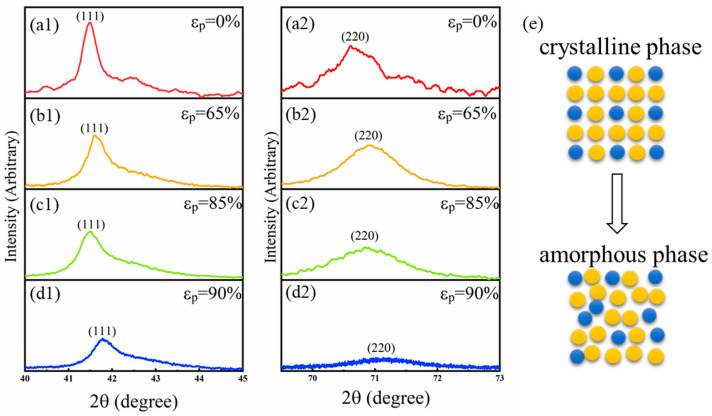
XRD patterns with two characteristic peaks ((**a1**–**d1**) (111), (**a2**–**d2**) (220) peaks) of FCC structure of the specimens with different thickness reduction *ε*_p_ ((**a**) original specimens, *ε*_p_ = 0%, (**b**) *ε*_p_ = 65%, (**c**) *ε*_p_ = 85%, and (**d**) *ε*_p_ = 90%)) at 300 K. (**e**) The schematics for the case where the crystalline phase is disrupted into the amorphous phase.

**Figure 2 materials-15-07900-f002:**
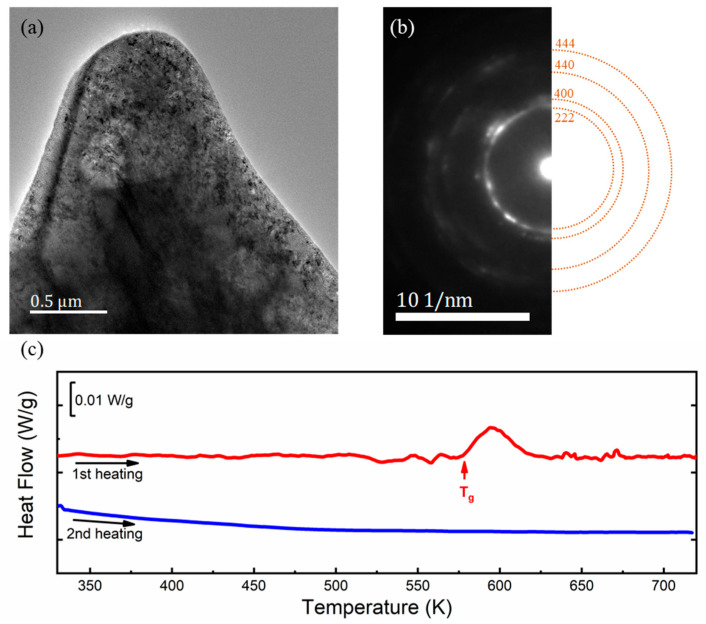
(**a**) TEM bright-field image and (**b**) selected-area diffraction pattern (SADP) reveal an amorphous state of the 90% cold-rolled specimen. The SADP is compared with that of the general FCC Fe_3_Pd crystal. (**c**) The DSC curves exhibit a glass transition from metallic glass to crystal, confirming that the crystallization temperature is around 580 K (the heating rate is 20 K/min).

**Figure 3 materials-15-07900-f003:**
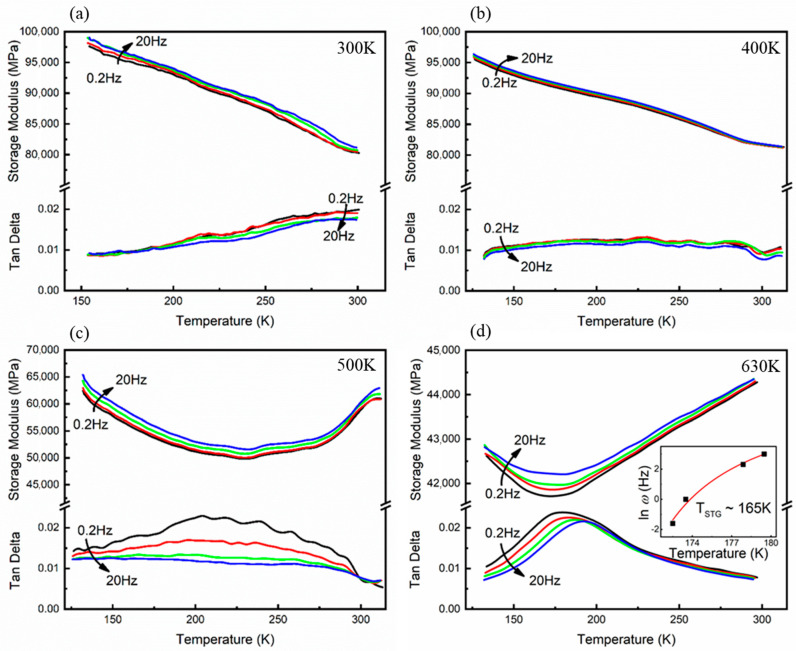
Elastic moduli (Storage Modulus) and internal friction (Tan Delta) curves of the specimens after (**a**) 90% cold rolling and then heating to (**b**) 400 K, (**c**) 500 K, and (**d**) 630 K for 1 h (the cooling rate is 2 K/min at AC field frequency 0.2 Hz, 1 Hz, 10 Hz and 20 Hz).

**Figure 4 materials-15-07900-f004:**
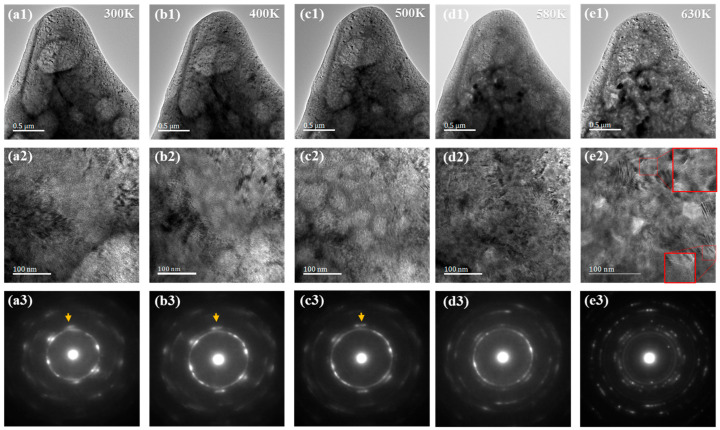
The TEM micrographs and SADPs of the 90% rolled specimens at (**a1**–**a3**) room temperature of 300 K and after heating to (**b1**–**b3**) 400 K, (**c1**–**c3**) 500 K, (**d1**–**d3**) 580 K, and finally (**e1**–**e3**) annealing at 630 K for 1 h. The middle images are the top images at a larger magnification and also selected areas for the bottom SADPs.

**Figure 5 materials-15-07900-f005:**
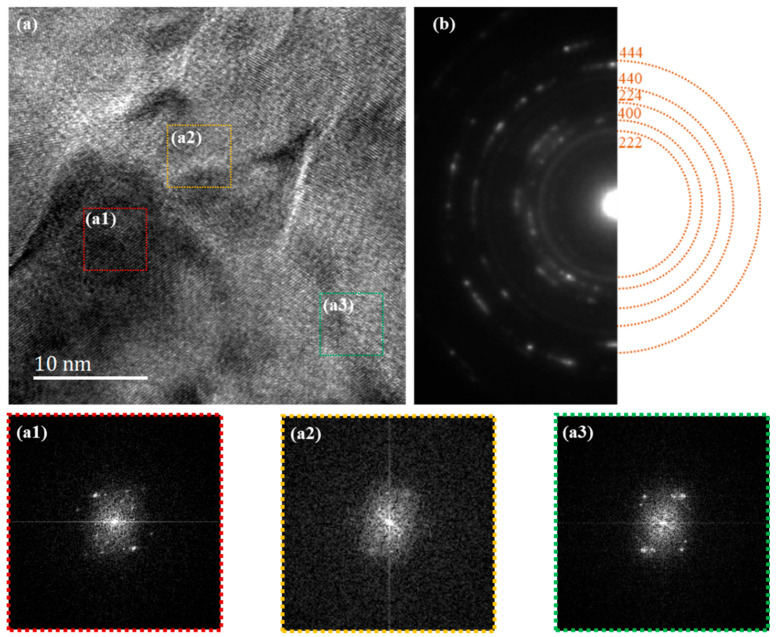
(**a**) The high-resolution TEM micrographs of the 90% rolled specimens annealed at 630 K for 1 h with the FFT diffraction patterns of the regions a1 (nanocrystal), a2 (amorphous), and a3 (nanocrystal), and (**b**) SADPs compared with that of general Fe_3_Pd FCC crystal.

**Figure 6 materials-15-07900-f006:**
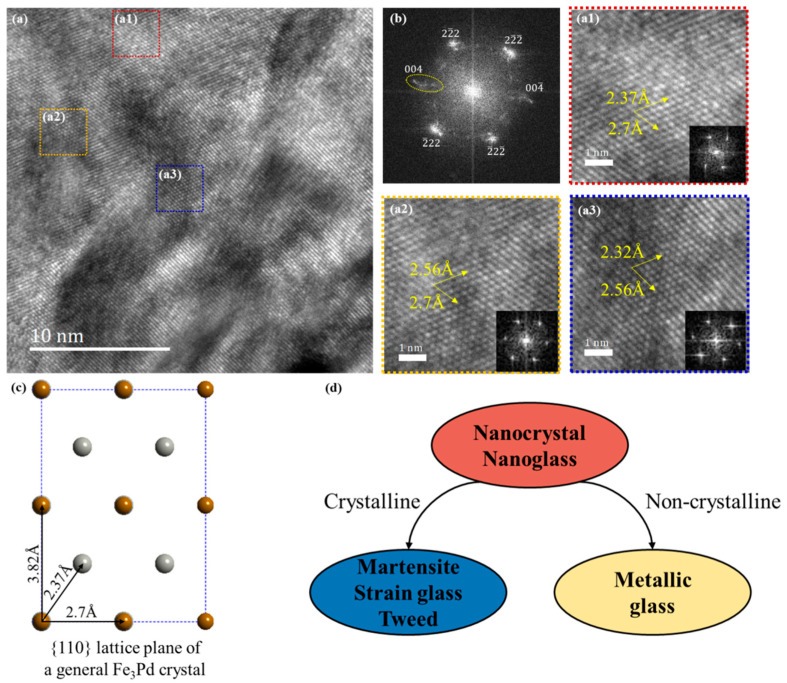
(**a**) The high-resolution TEM micrographs of the 90% rolled specimens annealed at 680 K for 1 h with (**b**) the FFT diffraction patterns of the whole region. The HRTEM of the regions a1, a2, and a3 are enlarged to measure lattice parameters with their FFT diffraction patterns. (**c**) Schematic picture of the {110} lattice planes. (**d**) Schematic picture of the evolution process.

**Table 1 materials-15-07900-t001:** Structure parameters of recrystallized alloy.

HRTEM Region	{110} Plane Atomic Spacing (Å)		*a* (Å)	*b* (Å)	*c* (Å)	Lattice
General Fe_3_Pd	2.7	2.37	3.818	3.818	3.818	FCC
a1	2.7	2.37	3.818	3.818	3.818	FCC
a2	2.7	2.56	3.818	3.818	4.350	FCT
a3	2.7	2.32	3.818	3.818	3.774	FCT

## Data Availability

The data that support the findings of this study are available within this article.

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
