# Peer review of "Direct Observation of Evolution from Amorphous Phase to Strain Glass"

_materials, 2022, doi:10.3390/ma15227900_

Round 1
Reviewer 1 Report
The topic covered in this manuscript is relevant to the field of metallic glasses, phase transition, and in-situ electron microscopy. Direct observation of phase transition is extremely important in understanding the pillars of materials science. A few comments/queries to help the readers of this paper:
1. The abstract should include a comment about the initial phase being "almost amorphous". Do mention the about high-doping and nucleation sites (e.g., manuscript lines 207-210), and which other industry/application relevant materials should be studied/tested with these techniques.
2. Need further details on the experimental procedure including, water quenching, sample size/weight used for DSC and DMA, and sample preparation details for TEM. These details are prerequisite for readers to follow/replicate the work, and increases citations to the paper.
3. Need further clarity on Figure 1 (d1) where a (111) XRD peak is observed. Kindly correlate the observance of this peak to Figure 2b, where some diffraction-like spots are seen on the amorphous rings of 222 and 444. These results seem to confirm the "almost amorphous" state of the sample hinted by the authors.
4. Suggest repeating DSC analysis in the 530-630K range with slower heating/cooling rates, ~5K/min. Slower measurements around recrystallization temperature help enhance the transition point and gives details. Since the DSC data is normalized, suggestion is to mention the sample weight.
5. For ease of reading and flow in the text, it is advised to move the manuscript lines 169-171 to at the end of manuscript line 163.
6. Highly recommend tabulating the crystal parameters from Section 3.3 and Figure 6. The values in the TEM image are not clearly readable. It will be easier for a reader to compare the standard and experimental values in a table, and for the paper to garner more citations.
Reviewer 2 Report
The article is about evolution from amorphous phase to strain glass. However, some issues must to be addressed:
- Abstract: Please start by expressing the aim of this paper, followed by the rest of the information. Typically, the abstract should provide a broad overview of the entire project, summarize the results, and present the implications of the research or what it adds to its field.
- The bibliographic foundation is important and well executed, however some new discussions should be inserted, authors should consider some new works (also from 2016 to 2022!!) in the literature, such as: DOI 10.3390/ma13040835.
- Please avoid bulk citation, like 1-3, 5-8 …
- The results are merely presented, not properly discussed. Please add explanations for the observed changes. Please give an extended discussion on the obtained results and correlate your findings with previous literature studies and prospective applications.
- More analysis and interpretation of the results should be added for a clearer understanding of observed experimental phenomena.
- The authors must to provide some details about importance of the research and their applicability.
- Please rewrite the conclusions in a more quantitative form and enhance the clarity of the conclusion section in order to highlight the results obtained.
- General check-up and correction of the English language is suggested. There are still some minor typos and grammatical errors.
The author needs to address the abovementioned points for the betterment of the manuscript.
Reviewer 3 Report
The paper entitled “Direct observation of evolution from amorphous phase to strain glass” is a novel work and is having interesting information for its publications. I can recommend this article in your esteemed journal. However, the authors have to address the following points.
1. In abstract, the authors have mentioned about amorphous sample preparation by 90% cold rolling which is not clear here. It has to be checked
2. Further, in the abstract, the sample preparation from arc melting is not mentioned. In addition, there is no specific outcomes through the present research work with appropriate reasons in the abstract. It has to be addressed.
3. In introduction, some more literature related to selection of alloy, arc melting and cold rolling are to be incorporated
4. The main objectives of the present work at the end of introduction are to be incorporated
5. Schematic diagram mentioning arc melting, cold rolling, characterization, and testing are to be incorporated in the experimental section. This will attract the readers
6. Why the authors have incorporate Pd of 32.2%? Is there any specific reasons? It has to be addressed
7. What is the temperature set in arc melting process? Further, there is no detailed information related to arc melting, and cold rolling processes (like make of company, power input, reduction ratio, etc…)
8. How much reduction was given in cold rolling? It is missing in the reduction. As mentioned in the abstract of 90%, this mean, it is very high. How many stages of rolling were carried out? All details are to be properly addressed
9. Some photograph of showing the prepared samples are to be incorporated
10. Have the authors observed any crack during 90% reduction in rolling?
11. To avoid cracking, have the authors used any lubricant?
12. From Fig.2b of SAD patterns, which is Fe, and Pd rings?
Round 2
Reviewer 2 Report
Correct citation is :
Vizureanu,P.; Nabialek, M.; Sandu, A.V.; B. Jez; Investigation into the Effect of Thermal Treatment on the Obtaining of Magnetic 340 Phases: Fe5Y, Fe23B6, Y2Fe14B and αFe within the Amorphous Matrix of Rapidly-Quenched Fe61+xCo10−xW1Y8B20 Alloys (Where x 341 = 0, 1 or 2). Mater. 2020, 13, 835.
instead of:
Petrica,V.; Marcin, N.; Andrei V.S.;Bartłomiej J. Investigation into the Effect of Thermal Treatment on the Obtaining of Magnetic 340 Phases: Fe5Y, Fe23B6, Y2Fe14B and αFe within the Amorphous Matrix of Rapidly-Quenched Fe61+xCo10−xW1Y8B20 Alloys (Where x 341 = 0, 1 or 2). Mater. 2020, 13, 835.
Author Response
We thank the reviewer for the constructive comments and suggestions that will surely help us improve our paper. The metioned citation has been corrected in the referrence part of the revised manuscript.
Changes made (Page 11 line 344):
“11. Vizureanu, P.; Nabialek, M.; Sandu, A.V.; B. Jez. Investigation into the Effect of Thermal Treatment on the Obtaining of Magnetic Phases: Fe5Y, Fe23B6, Y2Fe14B and αFe within the Amorphous Matrix of Rapidly‐Quenched Fe61+xCo10−xW1Y8B20 Alloys (Where x = 0, 1 or 2). Mater. 2020, 13, 835.”